# ROSA: RANDOM ORTHOGONAL SUBSPACE ADAPTERS FOR EFFICIENT FINE-TUNING

## ABSTRACT

Model training requires significantly more memory, compared with inference. Parameter efficient fine-tuning (PEFT) methods provide a means of adapting large models to downstream tasks using less memory. However, existing methods such as adapters, prompt tuning or low-rank adaptation (LoRA) either introduce latency overhead at inference time or achieve subpar downstream performance compared with full fine-tuning. In this work we propose Random Orthogonal Subspace Adapter (ROSA), a method that outperforms previous PEFT methods by a significant margin, while maintaining a zero latency overhead during inference time. In contrast to previous methods, ROSA is able to adapt subspaces of arbitrarily large dimension. We demonstrate both theoretically and experimentally that this makes ROSA strictly more expressive than LoRA, without consuming additional memory during runtime. As PEFT methods are especially useful in the natural language processing domain, where models operate on scales that make full fine-tuning very expensive, we evaluate ROSA in two common NLP scenarios: natural language generation (NLG) and natural language understanding (NLU) with GPT-2 and RoBERTa, respectively. We show that on almost every GLUE task ROSA outperforms LoRA by a significant margin, while also outperforming LoRA on NLG tasks. Our code will be made publicly available on acceptance.

## 1 INTRODUCTION

The advent of large language models pre-trained on web-size corpora (pre-trained LMs, or PLMs) has led to remarkably performant models in the domain of natural language processing (Brown et al., 2020; Devlin et al., 2019). As the size of such models ranges from hundreds of millions to hundreds of billions of parameters (Touvron et al., 2023), adapting them to downstream tasks is challenging and computationally expensive Peng et al. (2023). Compared to inference, training requires substantially more memory (2x-4x as much). For example, the GPT-2 base model (128M parameters) together with an input batch of 8 sequences of length 512 requires 670MB and 1139MB during inference and training, respectively (Radford et al., 2019).

To alleviate the burdensome memory requirements of adapting PLMs to downstream tasks, various memory efficient methods have been proposed (Houlsby et al., 2019; Lin et al., 2020; Guo et al., 2021; Hu et al., 2022; Li & Liang, 2021; Lester et al., 2021; Sung et al., 2021; Liu et al., 2022b;a). The commonality among these methods is the maintenance of fixed PLM weights while introducing a minimal quantity of trainable parameters. Although solutions like LoRA (Hu et al., 2022) and $(IA)^3$ (Liu et al., 2022a) are effective and do not impose any additional inference latency, they implicitly limit the expressivity of adapted models. For instance, LoRA adapts low-rank matrices that are added in parallel to fixed pre-trained weights. While this approach makes it possible to fine-tune large PLMs with reduced memory footprint compared to full fine tuning, it introduces an unavoidable bias: the pre-trained weight matrices can only be fine-tuned to matrices that are "a low-rank matrix away" from the initial weights.

In this work, we propose ROSA: **R**andom **O**rthogonal **S**ubspace **A**dapter, which expands the expressivity of adapted models, while remaining as memory efficient as LoRA. Similarly to LoRA, ROSA satisfies memory constraints by selectively fine-tuning low-rank matrices in parallel to fixed pre-trained weight matrices. Thus allowing users to fine-tune models in resource constrained settings. At the same time, ROSA alleviates the expressivity limitation of LoRA by continuously *sampling*

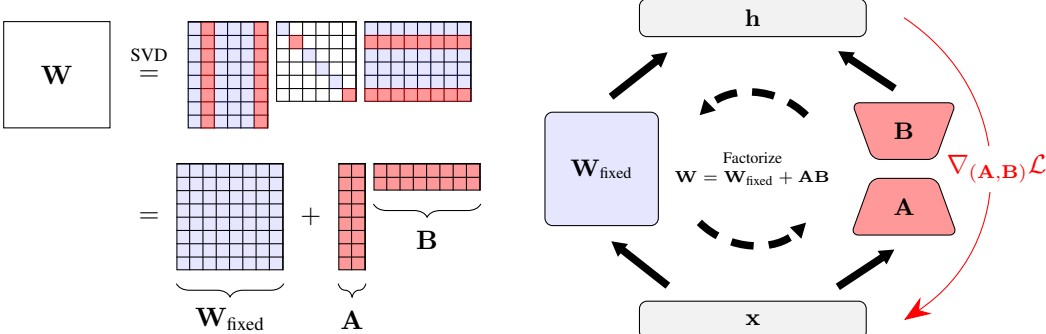

Figure 1: Illustration of ROSA. Parameter matrix $\mathbf{W}$ is factorized using the singular value decomposition (SVD) and split into smaller trainable matrices $(\mathbf{A}, \mathbf{B})$ and a larger fixed matrix $(\mathbf{W}_{\text{fixed}})$. Gradients during back-propagation are only computed with respect to $(\mathbf{A}, \mathbf{B})$. The split is then merged after a specified number of training iterations, and the process is repeated. ROSA updates an increasingly larger subspace of $\mathbf{W}$ over the course of training while remaining memory efficient.

different low-rank trainable subspaces and iteratively *merging* learned information into fixed weights throughout fine-tuning, as depicted in Figure 1. From a theoretical perspective, we formally characterize the implicit low rank bias of LoRA, show how this bias can be detrimental even on a simple regression task, and demonstrate that ROSA does not suffer from this limitation (see Theorem 1). Even further, our results show that (i) ROSA can fine-tune pre-trained weights to arbitrary target weights (i.e. is as expressive as full fine-tuning), and (ii) while LoRA trades *expressivity* for lower memory requirements, ROSA instead trades *convergence speed* for the same. These results are clearly and intuitively illustrated on a simple synthetic experiment presented in Section 4.1.

From a practical perspective, we show that ROSA achieves performance on par with full fine-tuning and consistently outperforms state-of-the-art methods such as LoRA and $(\mathsf{IA})^3$ (Liu et al., 2022a) on natural language understanding (GLUE) (Wang et al., 2019b) and natural language generation (E2E) (Novikova et al., 2017) tasks by significant margins. Lastly, we note that ROSA carries a significant advantage over approaches such as adapters (Houlsby et al., 2019) and prompt tuning (Li & Liang, 2021; Lester et al., 2021), as it introduces no additional latency overhead during inference time.

In summary, our key contributions are:

- Demonstrating both empirically and theoretically, that the low rank nature of LoRA can detrimentally limit its expressiveness.
- Introducing ROSA, a PEFT method that circumvents the low rank limitation of LoRA while remaining as memory efficient as LoRA.
- Proving that ROSA is more expressive than LoRA and can be as expressive as full fine-tuning.
- Conducting extensive experiments showing that ROSA consistently outperforms LoRA by a significant margin on natural language understanding (GLUE) and on natural language generation (E2E) benchmarks.

## 2    RELATED WORK

PEFT defines a class of methods to alleviate memory and compute requirements during adaptation of large models to downstream tasks by tuning only a relatively small number of added parameters, rather than the tuning all the parameters of the model itself.

**Adapters:**    Adapter methods such as in Houlsby et al. (2019) inject layers in between certain modules of each transformer block. These layers have relatively few parameters, as they project the original features down into a smaller set of dimensions, then scale them back up after applying the adapter's feed-forward layer. This structure necessarily leads to a latency overhead.

**Prompt and prefix tuning:** Prompt and prefix tuning are an efficient means of adapting models via continuous optimization of prefixes added to input prompts (Li & Liang, 2021; Lester et al., 2021; Liu et al., 2022b). While such approaches are memory efficient, they require reserving a portion of the available sequence length during downstream adaptation. Moreover, prompt tuning methods can be challenging to optimize as pointed out in Hu et al. (2022).

**LoRA:** Our work is most similar to LoRA (Hu et al., 2022), which has been shown to outperform the aforementioned approaches and to mitigate limitations such as increased inference latency and reduced sequence length capacity. LoRA adds a trainable low rank matrix to the frozen original weight matrix. The low rank matrix, parameterized as the product of two small matrices, is then fine-tuned instead of the original model weights. The authors of LoRA hypothesize that during task-specific fine-tuning, the model weight updates have a low "intrinsic dimension" and thus can be effectively approximated by low-rank matrices. While this may be true for some downstream tasks, we show both theoretically and empirically that this is not always the case and that restricting the weights update to a low intrinsic dimension can be detrimental.

**(IA)³:** Another widely known PEFT method is (IA)³ (Liu et al., 2022a). (IA)³ (Infused Adapter by Inhibiting and Amplifying Inner Activations) adds learned vectors to the attention and feedforward layers of the transformer, which are used to rescale the activations of these modules. (IA)³ further reduces the number of trainable parameters from LoRA, and makes the proportion of trainable parameters fixed, as the size of the rescaling vectors is directly dependent on the dimensions of the transformer's weight matrices.

**AdaLoRA:** Another competitive approach, AdaLoRA (Zhang et al., 2023) allocates a fixed parameter budget across the layers of a model dynamically by manipulating the rank, providing a lower rank to less important modules and vice versa. This is done via a novel importance metric that quantifies the contribution of a given module to the model's overall performance.

**Other methods:** BitFit (Ben Zaken et al., 2022) freezes all parameters except bias terms. FISH Sung et al. (2021) optimizes a sparse difference vector to be summed with the original model parameters.

## 3 METHOD

In this section we describe our proposed approach, ROSA: Random Orthogonal Subspace Adapter. The purpose of ROSA is to provide a means for fine-tuning large models in memory constrained settings while remaining competitive with full fine-tuning in terms of performance. After introducing ROSA and demonstrating its memory efficiency in Section 3.1, we provide a theoretical analysis showing that ROSA is provably more expressive than LoRA in Section 3.2.

### 3.1 ROSA

In LoRA (Hu et al., 2022), pre-trained models are adapted to alternative tasks by adding low rank matrices $\mathbf{A} \in \mathbb{R}^{M \times R}, \mathbf{B} \in \mathbb{R}^{R \times N}$ in parallel to pre-trained weights $\mathbf{W} \in \mathbb{R}^{M \times N}$. The output of the LoRA layer is given by

$$\phi(\mathbf{x}) = \mathbf{W}\mathbf{x} + \mathbf{A}\mathbf{B}\mathbf{x}. \tag{1}$$

The adapter weights $\mathbf{A}$ and $\mathbf{B}$ are initialized such that $\mathbf{AB} = \mathbf{0}$ and are the only parameters being updated during training. LoRA is memory efficient as $\mathbf{W}$ remains fixed during training (no gradient buffers necessary for the full weights) and typically $R \ll \text{MIN}(M, N)$. The rank $(R)$ of the trainable matrices is chosen such that the training procedure satisfies device memory constraints.

Constraining the updates to a *fixed* low rank subspace *initialized at zero* induces two limitations. First, the low rank nature of LoRA is such that the difference between the fine-tuned weight matrix $\mathbf{W} + \mathbf{AB}$ and the pre-trained weights $\mathbf{W}$ is constrained to be a low rank matrix. This significantly hinders the ability of LoRA to fine-tune a given model to an arbitrary target model/task. Note that even in the case where the target weights $\mathbf{W}^*$ are close to the pre-trained weights $\mathbf{W}$ (w.r.t. e.g., the Frobenius norm), this low-rank constraint creates an unavoidable bias when the difference $\mathbf{W}^* - \mathbf{W}$ is not low rank. We formally characterize this bias in Section 3.2 and empirically demonstrate it in Section 4.1. Second, initializing the adapter $\mathbf{AB}$ to zero can be thought of as learning new representations from

scratch separately from the pre-trained ones ($\phi(\mathbf{x}) = \mathbf{Wx} + \mathbf{ABx} := \phi_{\text{pre-trained}}(\mathbf{x}) + \phi_{\text{trainable}}(\mathbf{x})$), rather than leveraging the pre-trained features the model already has to initialize the adapter.

To address these limitations, ROSA continuously samples new weight subspaces to finetune throughout the training procedure. This offers several advantages. First, iteratively re-sampling new subspaces effectively expands the dimension of the fine-tuned subspace. Hence, ROSA does not suffer from the low rank bias of LoRA (which we theoretically show in Section 3.2). This allows us to decouple the adapted model's expressivity from device memory constraints. Second, by iteratively focusing the weight updates to a given randomly chosen weight subspace, we utilize the pre-trained knowledge to successively initialize the adapter weights to a different subspace of the pre-trained weights.

To achieve this, ROSA adapters successively factorize $\mathbf{W}$ into trainable and fixed weight subspaces using SVD. More precisely, let $\mathbf{W} = \mathbf{U\Sigma V}^\top$ be the SVD of $\mathbf{W}$, let $\mathbf{U}_R \in \mathbb{R}^{M \times R}$ be the matrix obtained by selecting a random subset of $R$ columns of $\mathbf{U}$ and let $\mathbf{\Sigma}_R \in \mathbb{R}^{R \times R}$ and $\mathbf{V}_R \in \mathbb{R}^{R \times N}$ denote the matrices obtained by selecting the same subset of singular values and right singular vectors. The ROSA factorization step is defined by

$$\mathbf{W} = \mathbf{W}_{\text{fixed}} + \mathbf{AB}, \quad \text{where } \mathbf{A} = \mathbf{U}_R\mathbf{\Sigma}_R \in \mathbb{R}^{M \times R}, \ \mathbf{B} = \mathbf{V}_R \in \mathbb{R}^{R \times N} \text{ and } \mathbf{W}_{\text{fixed}} = \mathbf{W} - \mathbf{AB}. \tag{2}$$

During training, gradients are computed only with respect to the $R$ dimensional subspace which consists of $R(M + N)$ parameters. In contrast, full fine-tuning requires optimizing $MN$ parameters. Thus, ROSA leads to a reduction in the number of trainable parameters, given by

$$\rho_{\text{train}} = \frac{MN}{R(M + N)}. \tag{3}$$

The factorization step is repeated throughout training at a pre-determined frequency (e.g., after each epoch of fine-tuning). The overall ROSA procedure is illustrated in Figure 1 and described in Algorithm 1. In practice, ROSA is applied simultaneously to all weight matrices of the model to fine-tune. While each subspace sampling step is expensive, $\mathcal{O}(\max(N, M)^3)$, it is only performed once every epoch. We show in the experiment section that the sample step adds negligible time to the training procedure in practice (see Table 1).

Since the weight matrix is factorized using SVD, the two resulting weight subspaces are orthogonal at initialization. Thus, each factorization step of ROSA can be seen as decomposing the corresponding layer's feature space into two orthogonal subspaces: a low-dimensional one to be fine tuned and a larger one which will be frozen until the next factorization step. More precisely, if we let $\phi(\mathbf{x}) = \mathbf{Wx}$ be the output of the layer to be fine-tuned using ROSA, the factorization step decomposes $\phi(\mathbf{x})$ in $\phi(\mathbf{x}) = \phi_{\text{fixed}}(\mathbf{x}) + \phi_{\text{trainable}}(\mathbf{x})$ where $\langle\phi_{\text{fixed}}(\mathbf{x}), \phi_{\text{trainable}}(\mathbf{x})\rangle = \langle\mathbf{W}_{\text{fixed}}\mathbf{x}, \mathbf{ABx}\rangle = 0$. The trainable feature map $\phi_{\text{trainable}}$ is then optimized until the next factorization step, while $\phi_{\text{fixed}}$ remains fixed. Note that the orthogonality is not enforced during fine-tuning, it is only satisfied at the initialization of the factorization step. Hence, ROSA can be seen as iteratively initializing adapters to random low dimensional projections of the full feature space and fine-tuning them.

## 3.2 THEORETICAL ANALYSIS

In this section we formally show how the low rank parameterization of LoRA limits its expressiveness and how ROSA circumvents this limitation.

First, It is easy to see that, by construction, the residual matrices obtained by fine-tuning weight matrices using LoRA are constrained to be low rank:

**Proposition 1.** *Let $\mathbf{W}_0$ be a weight matrix of a pre-trained model to be fine-tuned. Then, any fine-tuned weight matrix $\mathbf{W}_{LoRA}$ obtained using LoRA with rank parameter $R$ will be such that* $\text{rank}(\mathbf{W}_0 - \mathbf{W}_{LoRA}) \leq R$.

*Proof.* This directly follows from the fact that $\mathbf{W}_{\text{LoRA}} = \mathbf{W}_0 + \mathbf{AB}$ and $\text{rank}(\mathbf{AB}) \leq R$. $\square$

As a consequence, fine-tuning using LoRA suffers an unavoidable estimation bias which is not present in ROSA. In the following theorem, we (i) formally characterize this bias on a simple multivariate linear regression fine-tuning problem and (ii) derive a convergence rate of ROSA for linear regression demonstrating that it does not suffer from the same limitation.

---

**Algorithm 1** ROSA

---

**Input:** $\mathbf{W} \in \mathbb{R}^{M \times N}$, $R$ (desired rank), $K$ (factorization frequency), $\mathcal{L}$ (loss function)

1: $[\mathbf{A}, \mathbf{B}] \leftarrow [\mathbf{0}, \mathbf{0}]$
2: **for** $t = 1$ to $T$ **do**
3:     **if** $t \bmod K == 0$ **then**
4:         $\mathbf{W} \leftarrow \mathbf{W} + \mathbf{AB}$
5:         $\mathbf{U}, \mathbf{\Sigma}, \mathbf{V}^\top \leftarrow \text{SVD}(\mathbf{W})$
6:         $(i_1, \cdots, i_R) \leftarrow \text{RANDINTS}(R, \min(M, N))$
7:         $[\mathbf{A}, \mathbf{B}] \leftarrow [\mathbf{U}_{:,(i_1,\cdots,i_R)} \mathbf{\Sigma}_{(i_1,\cdots,i_R),(i_1,\cdots,i_R)}, \mathbf{V}^\top_{(i_1,\cdots,i_R),:}]$
8:         $\mathbf{W} \leftarrow \mathbf{W} - \mathbf{AB}$
9:     **end if**
10:    $[\mathbf{A}, \mathbf{B}] \leftarrow [\mathbf{A}, \mathbf{B}] - \nabla_{[\mathbf{A}, \mathbf{B}]} \mathcal{L}(\mathbf{W} + \mathbf{AB})$
11: **end for**

---

**Theorem 1.** *Consider a simple multivariate least-square regression problem:*

$$\arg\min_{\mathbf{W}} \|\mathbf{XW} - \mathbf{Y}\|_F^2$$

*where $\mathbf{X} \in \mathbb{R}^{n \times d}$ and $\mathbf{Y} \in \mathbb{R}^{n \times p}$ are the input and output data matrices, respectively. We assume that there exists a solution achieving zero error*[*].

*Consider the sequence of fine-tuned weight matrices obtained by ROSA with rank parameter $R$ starting from a pre-trained weight matrix $\mathbf{W}_0$, assuming that each intermediate minimization problem is solved exactly:*

$$\mathbf{W}_t = \mathbf{W}_{t-1} + \mathbf{A}_t \mathbf{B}_t \quad \text{where } \mathbf{A}_t, \mathbf{B}_t = \arg\min_{\mathbf{A} \in \mathbb{R}^{n \times R}, \mathbf{B} \in \mathbb{R}^{R \times n}} \|\mathbf{X}(\mathbf{W}_{t-1} + \mathbf{AB}) - \mathbf{Y}\|_F^2.$$

*Then, ROSA will converge to a fine-tuned matrix achieving zero error in at most*

$$T = \left\lceil \frac{\text{rank}(\mathbf{XW}_0 - \mathbf{Y})}{R} \right\rceil$$

*steps. That is, $\|\mathbf{XW}_t - \mathbf{Y}\|_F^2 = 0$ as soon as $t \geq T$.*

*In contrast, the error achieved by LoRA with rank parameter $R$ is lower bounded as*

$$\|\mathbf{XW}_{LoRA} - \mathbf{Y}\|_F^2 \geq \sum_{i=R+1}^{\min(d,p)} \sigma_i(\mathbf{\Pi_X Y} - \mathbf{XW}_0)^2$$

*where $\sigma_i(\mathbf{M})$ denotes the $i$th singular value of a matrix $\mathbf{M}$ (ordered decreasingly) and $\mathbf{\Pi_X}$ is the matrix of the orthogonal projection onto the range of $\mathbf{X}$.*

*Proof.* See Appendix A. □

Several observations are in order. First, Theorem 1 shows that even with rank parameter $R = 1$, ROSA will converge to the optimal solution of the linear regression fine-tuning problem (assuming that ROSA exactly solves the minimization problem between each factorization step). Second, increasing the rank parameter will lead to faster convergence. This suggests that the rank parameter $R$ in ROSA controls a trade-off between memory requirement and convergence speed. This is in stark contrast to LoRA, where the rank parameter controls the trade-off between memory requirement and expressiveness, as demonstrated in the previous theorem.

In the next section, we empirically demonstrate that this theoretical result also holds in practice when using LoRA and ROSA to fine-tune non-linear models trained using gradient based methods.

---

[*]This is only to simplify the theorem's statement. In the appendix we show a more general version of this theorem without this assumption

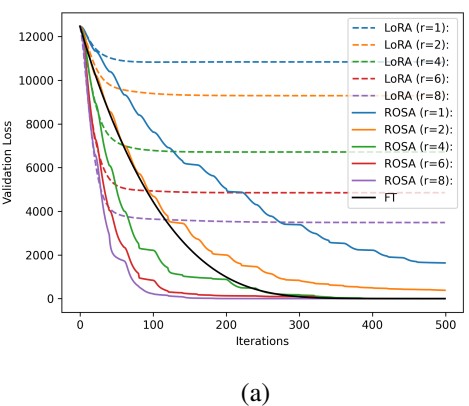 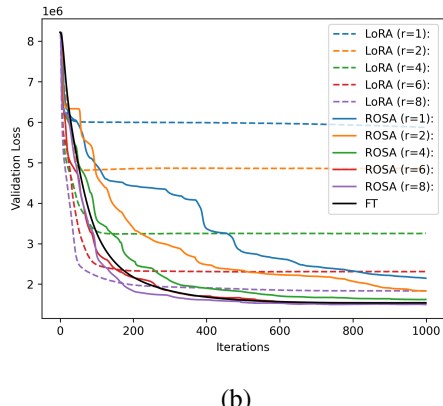

(a)                                              (b)

Figure 2: Validation loss curves of the training procedure for (a) 1-layer MLP and (b) 2-layer MLP (with ReLU activation). Both models are trained and evaluated on synthetic data that is generated from a randomly initialized MLP and a random low-rank adapter of rank 24. The models are trained to fit the synthetic data using the mean squared error loss function. This figure demonstrates that ROSA can find solutions with similar performance to full fine-tuning (FT) in practice.

## 4 EXPERIMENTS

In this section we compare the downstream performance of a model adapted using ROSA compared with LoRA (Hu et al., 2022) and $(IA)^3$ (Liu et al., 2022a) (as all three methods add zero latency overhead at inference time). For better comparison, in all experiments we use our own implementation for LoRA and $(IA)^3$ (detailed in Appendix B). In Section 4.1 we evaluate the performance of MLP models adapted to synthetic data, while Sections 4.2 & 4.3 evaluate the performance of RoBERTa$_{base}$ (Liu et al., 2019) and GPT-2 (Radford et al., 2019) on the GLUE and E2E benchmarks, respectively. In our experiments using transformer models, we only apply the PEFT methods to the attention layers, following a similar approach to Hu et al. (2022).

### 4.1 SYNTHETIC DATA

We first design two simple regression experiments with synthetic data to validate that the increased expressiveness of ROSA, illustrated in Theorem 1 for linear models, indeed leads to better results when fine-tuning non-linear models via Stochastic Gradient Descent.

To generate the synthetic data, we start by randomly initializing an MLP model $f$. We then add low rank matrices (rank=24), which are also randomly initialized, in parallel to the weights of $f$. This gives us the true model $f^*$, which we want to approximate. The synthetic data $\mathcal{D} = \{(\mathbf{x}, \mathbf{y})\}^n$ is generated by sampling $\mathbf{x} \sim \mathcal{N}(\mathbf{0}, \sigma\mathbf{I})$ and $\mathbf{y} = f^*(\mathbf{x})$.

The results are summarized in Figure 2, where we compare the evolution of the validation loss of ROSA and LoRA for fine-tuning the original MLP model $f$ to the data $\mathcal{D}$ generated from the target task $f^*$. As observed in Figure 2, ROSA at different rank values finds solutions with similar performance to full fine-tuning. This demonstrates that even in a more practical setting than the one of Theorem 1(namely with non-linear models trained by gradient descent) ROSA can match the performances of full fine-tuning. Moreover, for both 1-layer and 2-layers MLPs, we see that the rank limitation of LoRA prevents it from fully adapting to the target task: while increasing the rank leads to better validation loss, the unavoidable low-rank bias is clearly demonstrated by the convergence to a sub-optimal loss. In contrast, ROSA always converges towards the optimal loss, even with rank parameter set to 1, and increasing the rank parameter leads to faster convergence to the optimal loss. Notably, using ROSA to adapt a two layer MLP containing a non-linearity recovers a model that well approximates the true model used to generate the data. This suggests that the formal result shown in Theorem 1 holds beyond the simple linear regression setting.

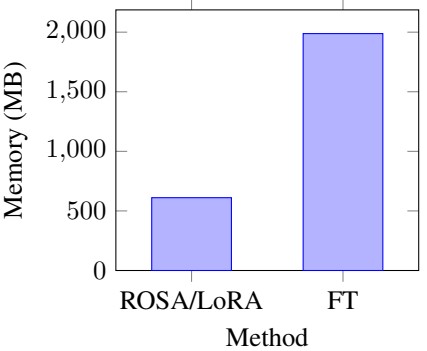

Figure 3: Memory usage during fine-tuning of RoBERTa$_{base}$ on the CoLA GLUE benchmark task, using ROSA compared with LoRA and full fine-tuning.

Table 1: Runtime of one epoch of fine-tuning of RoBERTa$_{base}$ (125M parameters) on the CoLA task, using ROSA and full fine-tuning. The experiment is conducted on a single GPU (NVIDIA A100-SXM4) with an input batch of 32 sequences of length 512. At every epoch a ROSA factorize step is performed, which adds negligible latency to the runtime during training.

| | Factorize Time (s) | Epoch Time (s) |
|---|---|---|
| FT | - | $157_{\pm 0.16}$ |
| LoRA | - | $152_{\pm 0.56}$ |
| ROSA | $4.03_{\pm 3.67}$ | $153_{\pm 0.16}$ |

Table 2: Performance of RoBERTa$_{base}$ fine-tuned using various PEFT methods (ROSA, LoRA and $(IA)^3$) on the GLUE benchmark tasks. We report the matched validation accuracy for MNLI, Matthew's correlation coefficient for CoLA, Pearson correlation for STS-B and accuracy for all other tasks.

| Method | Params | CoLA | MRPC | QNLI | RTE | STS-B | MNLI | SST2 | BoolQ | QQP |
|---|---|---|---|---|---|---|---|---|---|---|
| FT | 125 | 63.50 | 89.95 | 92.60 | 77.62 | 90.69 | 86.296 | 94.38 | 82.11 | 91.81 |
| $(IA)^3$ | 0.1 | 55.18 | 87.50 | 82.52 | 68.59 | 88.41 | 77.06 | 92.55 | 73.85 | |
| LoRA (r=2) | 0.2 | 53.42 | 88.72 | 92.10 | 72.56 | 83.91 | 85.37 | **93.69** | 72.96 | 88.97 |
| ROSA (r=2) | 0.2 | **62.08** | **89.46** | 92.20 | **73.64** | **89.32** | **86.25** | 92.55 | **76.78** | 89.09 |
| LoRA (r=8) | 0.6 | 54.27 | 88.24 | 92.20 | 69.31 | 82.10 | 85.88 | **93.57** | 68.37 | |
| ROSA (r=8) | 0.6 | **64.80** | 88.73 | **92.80** | 72.56 | **90.11** | **87.09** | 93.11 | **77.31** | 89.72 |

## 4.2 GLUE EXPERIMENTS

In this section we compare ROSA against LoRA and $(IA)^3$, by adapting RoBERTa$_{base}$ (125M) (Liu et al., 2019) on various tasks taken from the GLUE and SuperGLUE natural language understanding benchmarks (Wang et al., 2019b;a). These benchmarks cover a wide variety of natural language understanding tasks, including logical entailment, grammatical acceptability, question-answering, textual similarity, and sentiment analysis. The pre-trained model weights for RoBERTa$_{base}$ are taken from the Huggingface library (Wolf et al., 2020). A description of the specific subset of GLUE and SuperGLUE tasks tested on is available in Appendix B. Unlike some previous works which initialize the weights for MRPC, RTE, and STS-B with fine-tuned MNLI task-specific weights, we initialize the weights for all tasks with only the pre-trained RoBERTa weights for a fair comparison.

These tasks were selected to give a broad overview of ROSA's performance across a variety of different natural language tasks. We report development set performance for all tasks.

In Table 2 we show that ROSA outperforms LoRA and $(IA)^3$ by a significant margin on multiple tasks. Most notably, on CoLA using rank equal to eight we obtain Matthew's correlation coefficients of **64.80** for ROSA, **54.27** for LoRA and **55.18** for $(IA)^3$. Furthermore, ROSA remains as memory efficient as LoRA (Figure 3), and the factorization steps in ROSA add negligible latency (Figure 1). (See Appendix for training curves of RoBERTa$_{base}$ fine-tuned on CoLA for 10 epochs.)

## 4.3 NLG EXPERIMENTS

In this section we investigate the performance of ROSA in the natural language generation (NLG) setting. Namely, we compare the performance of GPT-2 Radford et al. (2019) using ROSA compared with LoRA and $(IA)^3$, when finetuned on the E2E NLG task Novikova et al. (2017). The E2E NLG

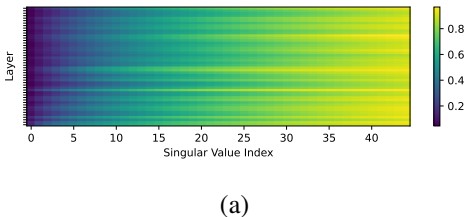 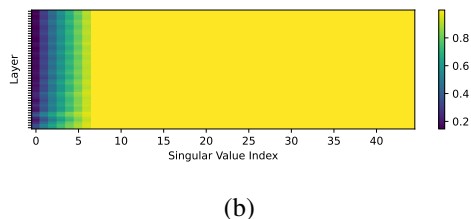

(a)                                                    (b)

Figure 4: Plot of the cumulative sum of singular values of residual matrices. The residual matrices in these plots are obtained from fine-tuning RoBERTa_base on CoLA for 10 epochs and achieving a Matthew's Correlation score of **(a)** 64.80 with ROSA and **(b)** 54.27 with LoRA. The figure demonstrates that the rank of the residual matrices obtained when adapting the RoBERTa_base using ROSA, is indeed far greater than the rank of the residuals obtained when fine-tuning using LoRA.

Table 3: Performance of GPT-2 finetuned on the End-to-End natural language generation task (E2E) Novikova et al. (2017), using ROSA, LoRA and $(IA)^3$.

| Name | # Trainable Parameters (M) | BLEU |
|---|---|---|
| FT | 355 | 68 |
| $(IA)^3$ | 0.2 | 65 |
| LoRA (r=4) | 0.9 | 64 |
| ROSA (r=4) | 0.9 | **68** |
| LoRA (r=8) | 1.7 | 67 |
| ROSA (r=8) | 1.7 | 67 |

Table 4: Performance of RoBERTa_base using variants of ROSA $_{r=8}$, finetuned on the CoLA and STS-B GLUE benchmark tasks.

| Name | CoLA | STS-B |
|---|---|---|
| FT | 63.52 | 90.69 |
| LoRA(Random Init) | 54.27 | 82.10 |
| ROSA(SVD Init) | 57.08 | 89.19 |
| ROSA(SVD Init + Ortho) | 60.32 | 89.42 |
| ROSA(SVD Init + Ortho + Resampling) | **64.80** | **90.11** |

task involves producing a fluent natural language description of a restaurant given a logical form describing its various attributes. The model's generations are compared against multiple reference texts to account for variations in wording. The score provided is the maximum BLEU score across all the reference texts for a given input.

In Table 3 we show that ROSA outperforms LoRA and $(IA)^3$ by a significant margin in the BLEU score.

### 4.4 WHAT COMPONENTS OF ROSA LEADS TO ITS PERFORMANCE?

In this section we empirically study several aspects of ROSA. We highlight three key components of ROSA, on which we perform ablation studies. The key components of ROSA are:

- **SVD Initialization:** ROSA adapters are initialized using SVD, as opposed to the random initialization of LoRA.

- **Orthogonality:** In ROSA, pre-trained weight matrices are decomposed such that the trainable adapter weights are initially *orthogonal* to the fixed weights.

- **Resampling:** In ROSA the difference between the pre-trained weights and the final weights is not constrained to be low-rank, due to resampling and merging of subpsaces throughout training.

We study the effects of progressively adding these components to ROSA in Table 4. We find that the progressive addition of the aforementioned components to ROSA is beneficial to its performance. The most drastic improvement is seen when resampling is added. This observation aligns with our theoretical analysis, demonstrating that the rank flexibility of ROSA, which increases the expressiveness of adapted models, leads to improved performance. Note also that the gain from LoRA to ROSA (SVD Init) aligns with our intuition that leveraging the pre-trained features to initialize

Table 5: Performance of RoBERTa$_{base}$ using different sampling schemes for ROSA, on the CoLA GLUE benchmark task. Top/Bottom/Random sampling indicate the method used for selecting the singular vectors to initialize the trainable subspace.

| Name | # Trainable Params | MRPC |
|---|---|---|
| FT | 125 | 63.52 |
| ROSA $_{r=8}$ (Top) | 0.6 | 58.88 |
| ROSA $_{r=8}$ (Bottom) | 0.6 | **60.57** |
| ROSA $_{r=8}$ (Random) | 0.6 | **60.32** |

the adapter is beneficial, compared to initializing the adapter to zero and learning new features from scratch.

We further investigate the rank structure of the matrices of the most performant RoBERTA$_{base}$ that achieved **64.80** on CoLA (with resampling). Specifically, we plot the singular values of residual matrices (defined as the difference between the initial pre-trained weights and the final weights upon completion of fine-tuning) in Figure 4. As shown in the figure, the ranks of the difference matrices achieved using ROSA (Figure 4a) are significantly larger than than the ranks of the difference matrices achieved using LoRA (Figure 4b).

### 4.5 INVESTIGATING DIFFERENT LOW-RANK SUBSPACE SAMPLING SCHEMES

In this section, we compare the random subspace sampling of ROSA with two other subspace selection strategies: selecting the top-$R$ or bottom-$R$ singular vectors . In doing so, we validate that performing random selection of singular vectors is as performant as selection based on singular value information. In Table 5, we report the performance of RoBERTa$_{base}$ fine-tuned on CoLA using the different sampling strategies, which confirm that, on this task, random sampling performs similarly or better than other schemes.

### 4.6 LIMITATIONS OF ROSA

While ROSA achieves better performance than previous state-of-the-art adaptation methods such as LoRA and (IA)$^3$, it bears one main limitation compared with other methods. Namely, it requires storage of the whole model after it is adapted for a downstream task.

Other adapter methods try to simultaneously address two challenges (1) reducing memory usage during training to ease the hardware barrier when adapting large models to a *single* downstream task and (2) reducing disk space usage when adapting a base model to *many* downstream tasks.

ROSA primarily focuses on addressing point (1), making it more suitable for scenarios involving a single downstream task. In comparison, other PEFT methods are better suited for scenarios involving multiple downstream tasks. ROSA excels in its specific domain, offering the same level of expressivity as full fine-tuning while requiring less GPU memory. This eliminates the need for (1) layerwise training, which would prolong training time, and (2) model sharding that necessitates more GPUs, thereby increasing training costs.

### 4.7 CONCLUSION & FUTURE WORK

In this work we introduced ROSA: Random Orthogonal Subspace Adapters. We first showed both theoretically and empirically that the low-rank nature of LoRA can often detrimentally affect its performance. In contrast, we demonstrate that ROSA can theoretically achieve the same solution as full fine-tuning. Furthermore, we demonstrate that on synthetic data ROSA indeed converges the same solution as full fine-tuning when using gradient based optimization. We evaluated ROSA against LoRA and (IA)$^3$ on both natural language understanding and natural language generation tasks. Our experiments showed that ROSA achieved performance similar to full fine-tuning and outperformed other state-of-the-art methods such as LoRA and (IA)$^3$ by significant margins. As our analysis was limited to adapting linear layers present in transformer models, adapting the parameters of convolution operations is an area for future work.

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

## APPENDIX

## A    THEOREM PROOFS

You may include other additional sections here.

**Theorem 1.** *Consider a simple multivariate least-square regression problem:*

$$\arg\min_{\mathbf{W}} \|\mathbf{XW} - \mathbf{Y}\|_F^2$$

*where $\mathbf{X} \in \mathbb{R}^{n \times d}$ and $\mathbf{Y} \in \mathbb{R}^{n \times p}$ are the input and output data matrices, respectively.*

*Consider the sequence of fine-tuned weight matrices obtained by ROSA with rank parameter $R$ starting from a pre-trained weight matrix $\mathbf{W}_0$, assuming that each intermediate minimization problem is solved exactly:*

$$\mathbf{W}_t = \mathbf{W}_{t-1} + \mathbf{A}_t\mathbf{B}_t \quad \text{where } \mathbf{A}_t, \mathbf{B}_t = \arg\min_{\mathbf{A} \in \mathbb{R}^{n \times R}, \mathbf{B} \in \mathbb{R}^{R \times n}} \|\mathbf{X}(\mathbf{W}_{t-1} + \mathbf{AB}) - \mathbf{Y}\|_F^2.$$

*Then, ROSA will converge to a fine-tuned matrix achieving the optimal error in at most $T = \lceil \frac{\text{rank}(\mathbf{XW}_0 - \mathbf{Y})}{R} \rceil$ steps. That is, $\|\mathbf{XW}_t - \mathbf{Y}\|_F^2 = \|\mathbf{\Pi_X Y} - \mathbf{Y}\|_F^2$ as soon as $t \geq T$, where $\mathbf{\Pi_X}$ is the matrix of the orthogonal projection onto the range of $\mathbf{X}$.*

*In contrast, the error achieved by LoRA with rank parameter $R$ is lower bounded as*

$$\|\mathbf{XW}_{LoRA} - \mathbf{Y}\|_F^2 \geq \sum_{i=R+1}^{\min(d,p)} \sigma_i(\mathbf{\Pi_X Y} - \mathbf{XW}_0)^2$$

*where $\sigma_i$ denotes the $i$th singular value (ordered decreasingly).*

*Proof.* First observe that the minimization problem optimized by LoRA,

$$\arg\min_{\mathbf{A},\mathbf{B}} \|\mathbf{X}(\mathbf{W}_0 + \mathbf{AB}) - \mathbf{Y}\|_F^2$$

is an instance of the *Reduced Rank Regression* problem Izenman (1975)

$$\underset{\mathbf{A}\in\mathbb{R}^{n\times R},\mathbf{B}\in\mathbb{R}^{R\times n}}{\arg\min} \|\mathbf{XAB} - (\mathbf{Y} - \mathbf{XW}_0)\|_F^2$$

whose optimal solution satisfies

$$\mathbf{AB} = ((\mathbf{X}^\top\mathbf{X})^{-1}\mathbf{XY} - \mathbf{W}_0)\sum_{i=1}^{R}\mathbf{v}_i\mathbf{v}_i^\top \tag{4}$$

where the $\mathbf{v}_i$'s are the first $R$ right singular vectors of the matrix $(\mathbf{\Pi_X Y} - \mathbf{XW}_0)$ and $\mathbf{\Pi_X} = \mathbf{X}(\mathbf{X}^\top\mathbf{X})^{-1}\mathbf{X}^\top$. The cost of the solution computed by LoRA can thus be lower bounded by

$$\begin{aligned}
\|\mathbf{XW}_{\text{LoRA}} - \mathbf{Y}\|_F^2 &= \|\mathbf{XW}_{\text{LoRA}} - \mathbf{\Pi_X Y}\|_F^2 + \|\mathbf{\Pi_X Y} - \mathbf{Y}\|_F^2 \\
&\geq \|\mathbf{\Pi_X Y} - \mathbf{XW}_{\text{LoRA}}\|_F^2 \\
&\geq \|\mathbf{\Pi_X Y} - \mathbf{X}\left(\mathbf{W}_0 + ((\mathbf{X}^\top\mathbf{X})^{-1}\mathbf{XY} - \mathbf{W}_0)\sum_{i=1}^{R}\mathbf{v}_i\mathbf{v}_i^\top\right)\|_F^2 \\
&= \|(\mathbf{\Pi_X Y} - \mathbf{XW}_0)(\mathbf{I} - \sum_{i=1}^{R}\mathbf{v}_i\mathbf{v}_i^\top)\|_F^2 \\
&= \sum_{i=R+1}^{\min(d,p)}\sigma_i(\mathbf{\Pi_X Y} - \mathbf{XW}_0)^2
\end{aligned}$$

where we used the fact that $\langle\mathbf{XW}_{\text{LoRA}} - \mathbf{\Pi_X Y}, \mathbf{\Pi_X Y} - \mathbf{Y}\rangle = 0$ for the first equality, and the fact that $\mathbf{W}_{\text{LoRA}}$ is of the form $\mathbf{W}_0 + \mathbf{AB}$ for the second inequality. This shows the second part of the theorem.

For the first part of the theorem, first observe that $\mathbf{W}_1 = \mathbf{W}_0 + \mathbf{AB}$ where $\mathbf{AB}$ is the solution of the reduced rank regression problem defined in Eq. (4). Similarly, one can show that the solution for the second step of ROSA is given by

$$\mathbf{W}_2 = \mathbf{W}_1 + ((\mathbf{X}^\top\mathbf{X})^{-1}\mathbf{X}^\top\mathbf{Y} - \mathbf{W}_1)\sum_{i=1}^{R}\tilde{\mathbf{v}}_i\tilde{\mathbf{v}}_i^\top$$

where the $\tilde{\mathbf{v}}_i$ are the first $R$ right singular vectors of the matrix $(\mathbf{\Pi_X Y} - \mathbf{XW}_1)$. However, we have

$$\begin{aligned}
\mathbf{\Pi_X Y} - \mathbf{XW}_1 &= \mathbf{\Pi_X Y} - \mathbf{X}\left(\mathbf{W}_0 + ((\mathbf{X}^\top\mathbf{X})^{-1}\mathbf{XY} - \mathbf{W}_0)\sum_{i=1}^{R}\mathbf{v}_i\mathbf{v}_i^\top\right) \\
&= (\mathbf{\Pi_X Y} - \mathbf{XW}_0) - (\mathbf{\Pi_X Y} - \mathbf{XW}_0)\sum_{i=1}^{R}\mathbf{v}_i\mathbf{v}_i^\top \\
&= (\mathbf{\Pi_X Y} - \mathbf{XW}_0)\sum_{i=R+1}^{\min(d,p)}\mathbf{v}_i\mathbf{v}_i^\top
\end{aligned}$$

Hence the top $R$ right singular vectors of $(\mathbf{\Pi_X Y} - \mathbf{XW}_1)$ are equal to the right singular vectors $\mathbf{v}_{R+1}, \mathbf{v}_{R+2}, \cdots, \mathbf{v}_{2R}$ of the matrix $(\mathbf{\Pi_X Y} - \mathbf{XW}_0)$.

It follows, by recurrence, that

$$
\mathbf{W}_t = \mathbf{W}_{t-1} + ((\mathbf{X}^\top\mathbf{X})^{-1}\mathbf{X}^\top\mathbf{Y} - \mathbf{W}_{t-1}) \sum_{i=(t-1)R+1}^{tR} \mathbf{v}_i\mathbf{v}_i^\top
$$

$$
= \mathbf{W}_{t-1}(\mathbf{I} - \sum_{i=(t-1)R+1}^{tR} \mathbf{v}_i\mathbf{v}_i^\top) + (\mathbf{X}^\top\mathbf{X})^{-1}\mathbf{X}^\top\mathbf{Y} \sum_{i=(t-1)R+1}^{tR} \mathbf{v}_i\mathbf{v}_i^\top
$$

$$
= \mathbf{W}_0(\mathbf{I} - \sum_{i=1}^{tR} \mathbf{v}_i\mathbf{v}_i^\top) + (\mathbf{X}^\top\mathbf{X})^{-1}\mathbf{X}^\top\mathbf{Y} \sum_{i=1}^{tR} \mathbf{v}_i\mathbf{v}_i^\top
$$

$$
= \mathbf{W}_0 + ((\mathbf{X}^\top\mathbf{X})^{-1}\mathbf{X}^\top\mathbf{Y} - \mathbf{W}_0) \sum_{i=1}^{tR} \mathbf{v}_i\mathbf{v}_i^\top
$$

Hence, as soon as $t > \lceil \frac{\mathrm{rank}(\mathbf{X}\mathbf{W}_0 - \mathbf{Y})}{R} \rceil$, we have

$$
\mathbf{X}\mathbf{W}_t = \mathbf{X}\mathbf{W}_0 + (\mathbf{\Pi_X}\mathbf{Y} - \mathbf{X}\mathbf{W}_0) \sum_{i=1}^{tR} \mathbf{v}_i\mathbf{v}_i^\top = \mathbf{\Pi_X}\mathbf{Y}
$$

hence $\|\mathbf{X}\mathbf{W}_t - \mathbf{Y}\| = \|\mathbf{\Pi_X}\mathbf{Y} - \mathbf{Y}\|$, which concludes the proof. $\qquad\square$

## B  EXPERIMENTAL SETUP

### B.1  IMPLEMENTATION OF ROSA, LoRA AND (IA)$^3$

For better comparison, we re-implement the LoRA and (IA)$^3$ PEFT to share the same structure as ROSA. For all methods we use a vanilla implementation to focus on only comparing their core aspects. We list out the key differences between our implementation and those of LoRA Hu et al. (2022) and (IA)$^3$ Liu et al. (2022a).

- We apply adapters to all attention matrices. In contrast, LoRA tunes the attention matrices to which its adapter should be applied.
- We do not add additional dropout modules inside the adapter, as is done in the LoRA paper.
- We do not apply adapters to MLP layers as done so in (IA)$^3$.
- We use the same number of epochs across all model types (full fine-tuning, adapter). In contrast, LoRA and (IA)$^3$ experiments are typically run for far more epochs (roughly 3X) than the full fine-tuning experiments.

### B.2  GLUE EXPERIMENTS

For each experiment on GLUE we tune the LR for all three PEFT models for each selection of rank. Specifically, for a given task, model, PEFT method and rank value, we report the model that obtains the best validation set accuracy using LRs in $\{2e-2, 2e-3, 2e-4, 2e-5\}$. We use a factorization frequency of 2 in ROSA for all GLUE experiments (i.e., we merge then factorize the weight matrices once every two epochs.) We use the AdamW optimizer with $\beta_1, \beta_2 = (0.9, 0.98)$, $\epsilon = 1e-6$ and weight decay of $0.1$. Our batch size is selected from the set $\{16, 32\}$ and we use a sequence length of $512$. We train all models for 10 epochs.

Below is a description of each of the GLUE/SuperGLUE tasks selected for evaluation:

1. CoLA: a single-sentence classification task, where each sentence is labelled as either grammatical or not in English. The Matthews correlation coefficient is the reported metric.
2. MRPC: a sentence-pair classification task, where each pair of sentences is labelled as either semantically equivalent (i.e. paraphrases of each other), or not.
3. QNLI: QNLI is a sentence-pair classification task, where each pair of sentences corresponds to a paragraph from Wikipedia and a question, and the model must predict if the answer to the question is contained within the paragraph.

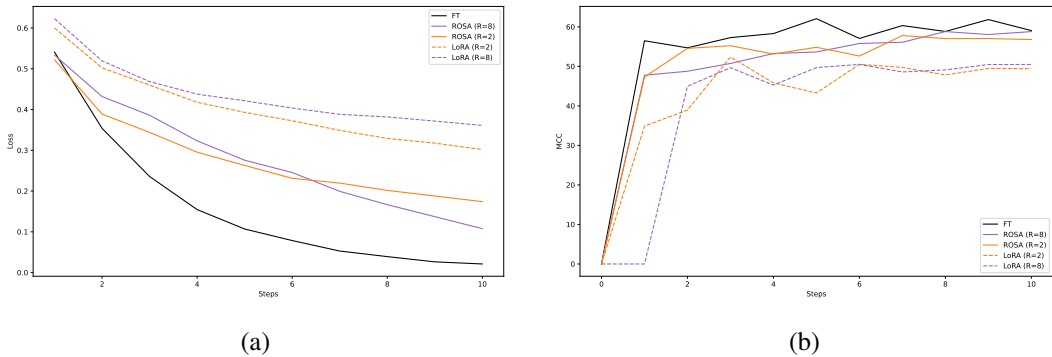

(a)  (b)

Figure 5: Plots of (a) Train loss and (b) and Validation Matthew's Correlation Coefficient. The plots are obtained from fine-tuning RoBERTa$_{base}$ on CoLA for 10 epochs.

4. RTE: The input for RTE is a pair of sentences, where the model must predict if the second sentence can logically be inferred from the first, or if it contradicts/is unrelated to it (binary classification).

5. STS-B: The only regression task in the GLUE Benchmark, for STS-B the model must predict the similarity between a pair of sentences on a continuous scale of 1 to 5. The reported metric is the Pearson correlation coefficient.

6. MNLI: The input for MNLI is a premise and a hypothesis, and the model must predict if the premise entails, contradicts, or is neutral toward the hypothesis. This is the same as RTE but with a separate neutral (unrelated) class. We report accuracy for the in-domain (matched) set.

7. SST2: SST2 is a binary (positive/negative) sentiment classification dataset of movie reviews.

8. BoolQ: BoolQ is a question-answering task where the input is a Wikipedia paragraph and a yes/no question where the answer is contained within the Wikipedia paragraph. The model must predict the answer to the question.

9. WiC (Words-in-Context): WiC is a binary sentence-pair classification task of disambiguating word senses. Two sentences are provided to the model that contain the same word,, and the model must predict if the same sense of the word is used in both cases.

### B.3  E2E NLG EXPERIMENTS

The E2E experiments are all carried out for 5 epochs. We also tune the LR in for each PEFT model by searching the space $\{2e-2, 2e-3, 2e-4, 2e-5\}$. We use the AdamW optimizer with $\beta_1, \beta_2 = (0.9, 0.999)$, $\epsilon = 1e-8$, weight decay of $0.1$, batch size of $10$ and a sequence length of $512$.

An example input and output for E2E is provided.

Input:    `name[The Vaults], eatType[pub], priceRange[more than £30], customer rating[5 out of 5], near[Café Adriatic]`

Output: *"The Vaults pub near Café Adriatic has a 5 star rating. Prices start at £30."*

### B.4  TRAINING CURVES FOR GLUE EXPERIMENTS

In Figure 5, we plot the training loss and validation curves for the fine-tuning of RoBERTa$_{base}$ on CoLA for 10 epochs.

