# OpenReview forum: "ROSA: Random Orthogonal Subspace Adaptation"
_ICLR.cc/2024/Conference — Submitted to ICLR 2024_

### Official Review · Reviewer_DBFm · 2023-10-30

**Soundness:** 3 good
**Presentation:** 3 good
**Contribution:** 3 good
**Rating:** 8
**Confidence:** 3

**Summary:**

This paper proposes Random Orthogonal Subspace Adapter (ROSA), a Parameter-efficient fine-tuning (PEFT) method that alleviates the expressivity limitation of previous solutions and does not incur extra latency overhead during inference. ROSA achieves this by iteratively decomposing weight matrices into low-rank trainable subspaces and orthogonal fixed subspaces and merging learned information. Their experiments show that ROSA outperforms LoRA on both GLUE and NLG tasks.

**Strengths:**

1. The paper observes and formally characterizes the expressivity limitation of SOTA PEFT method.
2. It sounds reasonable that the proposed ROSA method can expand the expressiveness.
3. ROSA does not incur large overhead for fine-tuning.

**Weaknesses:**

1. People are increasingly interested in LoRA than other PEFT methods because LoRA stores and loads a small number of task-specific parameters during inference. ROSA reintroduces this challenge, making it less practical than LoRA for inference.
2. The paper did not use real-world datasets to verify the expressive ability of ROSA. Considering that there are many real-world datasets suitable for regression experiments, there is no need to use synthetic data as in Section 4.1.

**Questions:**

Is it possible to perform some extra steps before merging to enforce orthogonality?

---

> ### Author Response · Authors · 2023-11-19
> **Response regarding limitation of ROSA**
>
> We would like to thank the reviewer for his feedback, pointing out the main limitation of ROSA compared to other PEFT methods. We have added an additional sentence in the Limitation section (4.6) to better indicate the settings in which ROSA excels.
>
> **Response to weakness 1**- We are aware this benefit of storing few parameters on disk when fine-tuning a baseline model for many downstream tasks is absent in ROSA. However, we think that there are many settings in which users would like to fine-tune a model for a single downstream task only. In such settings, ROSA shines as it provides the same benefits as LoRA while maintaining the same expressivity as full fine-tuning. We have added a few sentences to Section 4.6 to try emphasize the settings in which ROSA excels.
>
> **Response to weakness 2**- The intent of the synthetic experiments is to illustrate how ROSA bypasses the low rank limitation of LoRA in a simple controlled environment where we know exactly what the target rank is — with a real-world dataset, we would not know precisely what the target rank of the adapter is. We have also made an adjustment to the figure, adding the loss curve for full fine-tuning for better comparison. We have also added a few sentences to Section 4.1 to clarify the intent of this figure.
>
> **Response to question 1**- We are not entirely sure by what is meant by performing some extra steps before merging to enforce orthogonality. If you are asking if it is possible to include experiments where we actually enforce orthogonality before performing the merge operation, then this could indeed be another possible direction for performing the merge operation. However, this may reduce model performance as it would amount to merging a projection of the trained PEFT parameters.

---

> > ### Comment · Reviewer_DBFm · 2023-11-22
> >
> > Thank you for your response.

---

### Official Review · Reviewer_A1HC · 2023-11-01

**Soundness:** 3 good
**Presentation:** 3 good
**Contribution:** 2 fair
**Rating:** 5
**Confidence:** 3

**Summary:**

The paper introduces a new parameter-efficient adaptation scheme named ROSA: Random Orthogonal Subspace Adapters. The paper primarily targets the limited expressiveness of the existing PEFT methods and proposes a new adaptation scheme for increasing the expressiveness. The proposed fine-tuning strategy can be summarized as follows 1) For a pretrained model, low dimensional subspaces are obtained using SVD computation, 2) the optimization subspace is selected randomly, and the model is updated over the selected subspace for arbitrary iterations, 3) Further the pretrained model weights are again updated with the new weight approximations and new optimization subspace is again selected by considering random orthogonal subspaces obtained via SVD. The paper further shows that the proposed fine-tuning strategy is capable of fine-tuning pretrained weights to arbitrary target weights, making them as expressive as full-finetuning.

Apart from theoretical analysis, the paper provides a detailed set of experiments over widely used Natural language understanding and natural language generation benchmarks. The empirical results show a performance comparison with other PEFT approaches as well as the fine-tuned version of the model, highlighting the performance obtained by the proposed fine-tuning strategy to perform comparable/better to the full finetuning of the pretrained model.

**Strengths:**

* The paper provides a detailed theoretical analysis of the proposed scheme and shows the increase in expressiveness of the proposed architecture, highlighting the limitation when compared to the existing PEFT approaches (specifically LORA). The theoretical results are also backed up with detailed experimentation on various NLU and NLG tasks.
* The paper is well-written and compares the proposed method with LORA clearly. The primary components are the SVD initialization, orthogonality, and resampling. The paper also reports the ablation results, making the components justified and the study reliable for future research.
* The paper clearly talks about the limitations of the proposed method, which include the storage requirement of keeping the entire model after adaptation over a domain. The primary advantage of the proposed scheme is the reduction in memory usage during training, making the models trainable with limited GPU memory with the added limitation of being usefull for only a single downstream task.

**Weaknesses:**

* Since the entire model parameters are updated, it becomes crucial to consider a comparison with the full fine-tuning training strategy. If the convergence speed of the full-finetuning is not improved or comparable, the primary advantage of ROSA only relies on low memory usage for finetuning, which can also be achieved by other means like model sharding for faster training with a number of GPUs.

* The paper compares the training time of one epoch of fine-tuning in Table 1, where the Epoch time of full finetuning, as reported, is ~157 seconds. When comparing with the proposed method, the Epoch time + Factorize Time also results in a similar time of ~153 seconds + 4 seconds, since at every epoch, a ROSA factorization step is performed, the overall finetuning time of the proposed method and the finetuning of the entire model is similar. If the convergence speed of the proposed method is similar (or lower) to finetuning, the primary motivation of PEFT approaches (less training time) turns out to be missing. I am not sure if I am missing something; however, if the entire model is updated (not keeping the source knowledge intact in terms of pretrained model parameters as done in other adapter-based approaches), the proposed method results in the same model performance as finetuning with no advantage of PEFT approaches. Moreover, as highlighted by the authors, this also limits the usage of the proposed technique for multi-task/domain adaptation.

**Questions:**

* The proposed scheme seems like an Alt-Opt optimization scheme, where the model is updated on a few of the selected low-rank subspace directions, keeping other subspace directions fixed for a single update and later iterating over different directions to update the entire model’s parameter space. More detailed discussion on similar lines would help understand the proposed method better.

* A detailed analysis of the convergence rate with various PEFT methods, along with the full finetuning, would be required to obtain a transparent picture of the proposed PEFT technique. It would be interesting to observe if the proposed scheme helps converge with less variability and low sensitivity.

Minor suggestions:
* In Figure 2, it would be good to make a comparison with full-finetuning, highlighting the comparison between the rate of convergence.

---

> ### Author Response · Authors · 2023-11-19
> **Response regarding model sharding and emphasizing the benefit of ROSA**
>
> We thank the reviewer for his detailed feedback, pointing out improvements such as including training curves for better transparency and including the full fine-tuning baseline in Figure 2 for a clearer comparison between the PEFT methods with baseline.
>
> **Response to weakness 1**- Using model sharding for fine-tuning will split the total memory used to train a model across multiple GPUs. In cases where users have access to many GPUs to fine-tune their models for a single task, then there is no need to use any PEFT methods. However, in resources limited settings, users would like to fine-tune models using a limited number of GPUs with a collective memory that is less than the memory required for full fine-tuning of a model. In these settings PEFT models are needed as the only alternative would be layer-wise training. However, layer-wise training incurs significant latency overhead due to constant data movement onto and from the GPUs. We have added a discussion on this point in section 4.6
>
> **Response to weakness 2**-
> The primary advantage of PEFT methods is reducing the memory required to fine-tune a model on a downstream task, as opposed to speeding up training. The motivation is to allow users with potentially a single GPU to have the ability to fine-tune a large scale model in a memory constrained setting. This can be achieved by both LoRA and ROSA as they both provide a means of fine-tuning large models using significantly less memory. Moreover, other PEFT methods do not always lead to a speedup on GPUs (as shown in Table 1,  LoRA and Full fine-tuning all have comparable inference time). However, we realize the motivation may have been a little under-emphasized so we have reworded some sentences throughout the paper to try emphasize the motivation of ROSA (see text in blue in the introduction and at the beginning of the methodology section).
>
> **Response to question 1**-
> We are not very clear on this question. Do you mean that we should better explain in the paper that in ROSA the model is updated on a few of the selected low-rank subspace directions, while keeping other subspace directions fixed for a single update and later iterating over different directions?
>
> **Response to question 2**-
> We agree with you on this point. For better transparency we have added the training curve of CoLA to the paper in the appendix, and refered to it in the experimental section. Currently we do not observe a major differences between the convergence of ROSA, LoRA and full fine-tuning on the CoLA dataset. We plan to add more of the GLUE dataset training curves to the paper in the same manner.
>
> **Response to question 3**- Per your suggestion we have added a comparison with full fine-tuning in Figure 2

---

> > ### Comment · Reviewer_A1HC · 2023-11-22
> > **Response to the updated version**
> >
> > Thank you for your detailed response and updates to the paper.
> > Updating the paper with the training plots does improve the transparency of the paper.
> > There was some confusion regarding the Alt-Opt scheme mentioned in the questions.
> > The Alt-Opt in the question referred to the Alternating optimization scheme of updating the model parameters, which is discussed by you in the updated version of the paper, referred to as layer-wise training (also mentioned in the Response to weakness 1).
> >
> > The primary motivation behind the working of PEFT techniques like LoRA is that the change in weights during model adaptation has a low “intrinsic rank” since an overparameterized model resides in a low-rank subspace. [Li et al. (2018) and Aghajanyan et al. (2020)]. In ROSA, the entire model is updated by using multiple low-rank subspaces, increasing the overall intrinsic dimensions of the subspace used in fine-tuning (also observed in Figure 4 of the paper). Moreover, the learning curves depict a slower convergence rate. Since the proposed scheme is expressive as the entire model fine-tuning or layer-wise training, it would be better to show a comparison with these techniques as well, both in terms of performance as well as overall training time. These will not only help increase the reliability of the proposed method but also make it easier to decide between other memory reduction techniques like layer-wise fine-tuning/gradient accumulation, etc., based on the available computation budget and training time requirements.
> >
> > [1] Chunyuan Li, Heerad Farkhoor, Rosanne Liu, and Jason Yosinski. Measuring the Intrinsic Dimension of Objective Landscapes. arXiv:1804.08838 [cs, stat], April 2018.
> >
> > [2] Armen Aghajanyan, Luke Zettlemoyer, and Sonal Gupta. Intrinsic Dimensionality Explains the Effectiveness of Language Model Fine-Tuning. arXiv:2012.13255 [cs], December 2020.

---

> > > ### Author Response · Authors · 2023-11-22
> > > **Response to comparison with full fine-tuning and layerwise tuning**
> > >
> > > **Response to: "Since the proposed scheme is expressive as the entire model fine-tuning or layer-wise training, it would be better to show a comparison with these techniques as well, both in terms of performance as well as overall training time"**-
> > > We do indeed include a direct comparison with full fine-tuning in the paper both in terms of latency (Table 1) and performance (Table 2, Table 3, Table 4). For layer-wise training it is prohibitively expensive to compare against, as we found layer-wise training to be around **20x slower** than full fine-tuning due to the data movement. Here is the latency comparison on a smaller scale experiment
> > >
> > > ```Layer-wise training forward pass: 4.36ms ± 1.14ms (num samples = 1000)```
> > >
> > > ```Full fine-tuning forward pass: 0.18ms ± 0.01ms (num samples = 1000)```
> > >
> > > Since we've addressed your concerns regarding adding training plots for transparency and comparison with full fine-tuning as well as layer-wise training, would you consider raising your score?

---

> ### Author Response · Authors · 2023-11-21
> **Last day for the discussion period**
>
> To facilitate a discussion we encourage you to provide a response before the end of the discussion period (Nov. 22).

---

### Official Review · Reviewer_CdeU · 2023-11-09

**Soundness:** 2 fair
**Presentation:** 3 good
**Contribution:** 2 fair
**Rating:** 5
**Confidence:** 4

**Summary:**

This paper proposes a new finetuning method, called ROSA, for parameter-efficient fine-tuning. ROSA is short for random orthogonal subspace adapters that first factorize the parameter matrix W using singular value decomposition (SVD) and split it into smaller trainable matrices (A, B) and a larger fixed matrix (Wfixed). Gradients during back-propagation are calculated only with respect to (A, B). ROSA can maintain a low memory consumption in training as LoRA, while achieving better performance. The authors show that ROSA is better than LoRA theoretically and empirically. Experiments are performed with a few NLU and NLG tasks.

**Strengths:**

1. This paper proposes a new method to do parameter-efficient fine-tuning. The problem is important and the idea is interesting.

2. The authors carry out theoretical analysis and mathematical proof. They also provide empirical proof with NLU and NLG tasks.

3. This paper is well-written and easy to follow.

**Weaknesses:**

1. The ROSA method has been published in ICML23 [1]. The main idea is similar. This paper provides a more solid theoretical analysis and compares ROSA with LoRA on a few NLU and NLG tasks. However, improvements do not appear to be significant.

2. I think more experiments are needed to verify the effectiveness of the proposed method. (1) I'm not very clear about the task selections in the experiment section.  For example, the QQP task is missing in GLUE tasks. Ablation studies are performed in two different tasks (CoLA and MRPC). In my opinion, both tasks use a small training set and the results always fluctuate a lot. (2) PEFT methods are always used for large models. It is necessary to check the performance of large models. (3) LoRA is particularly useful for image generation tasks such as SD models. Would you do more experiments in computer vision tasks?

3. As discussed in the limitation section, LoRA can be released as a plunge into using very large models. It provides great flexibility in the use and distribution of models. Therefore, ROSA's practicability is weaker than LoRA, which will limit the contribution of this work.

[1] Gamal, Marawan, and Guillaume Rabusseau. "ROSA: Random Orthogonal Subspace Adaptation." Workshop on Efficient Systems for Foundation Models@ ICML2023. 2023.

**Questions:**

1. I do not quite understand of the limitation 2 of LoRA. The authors claim that "Second, initializing the adapter AB to zero can be thought of as learning new representations from scratch separately from the pre-trained ones (φ(x) = Wx + ABx:= φpre-trained(x) + φtrainable(x)), rather than leveraging the pre-trained features the model already has to initialize the adapter."

From my personal point of view, this design element is a boon rather than a limitation. It allows fine-tuning to start precisely from the pre-trained checkpoint, thus safeguarding the integrity of the pre-trained model. In the ROSA context, users do not need to initiate new parameters, as they can start training directly from the pre-trained checkpoint. Both methods are trained using pre-trained checkpoints. Therefore, I would not consider it a LoRA limitation. What are your thoughts?

2. A question about synthetic data experiments. As we know, LoRA is a method for finetuning, which means that the basic model is well-pre-trained. It assumes that the fine-ting tasks or domain adaptations can be achieved with a low-rank tuning. So I'm a bit confused about the synthetic data experiment. It can be shown that in some cases LoRA can not be optimized well. However, this is not the typical application of methods like LoRA.

---

> ### Author Response · Authors · 2023-11-19
> **Response regarding running additional experiments and clarifying the intent of synthetic experiments**
>
> We would like to thank the reviewer on their thorough feedback. Based on the reviewer's feedback we have ran additional experiments and included them in the revised PDF file as well as included clarifying remarks (highlighted in blue text).
>
> **Response to weakness 1** -
> The ROSA method was presented as a poster in a non-archival workshop very recently (less than 4 months from the submission deadline of ICLR). As mentioned in the guidelines, authors are not required to compare their work to other works not published in peer-reviewed conference proceedings or journals, as well as to contemporaneous work (less than 4 months before the ICLR deadline).
>
> **Response to weakness 2** -
> 1. We have added the QQP results for LoRA and ROSA in the updated PDF file for rank=2 (rank=8 results are pending). Moreover, to improve our ablation study - we are are currently running experiments on the STS-B dataset that will be completed by the end of the author-reviewer discussion period.
> 2. Unfortunately, we do not have access to industry scale computing resources, which prevents us from running experiments on very large language models. Note that the experiment results provided in the paper already required 100 GPU/Days of compute time. We also believe that the results in the paper already deliver strong evidence to ROSA’s efficiency.
> 3. Investigating the efficacy of our method in other domains such as computer vision is indeed an interesting direction. However, this goes beyond the scope of this paper and is deferred to future work.
>
> **Response to weakness 3** -
> Indeed, this is the main limitation of ROSA compared with LoRA. However, we believe there are many situations where there is a need to fine-tune a model efficiently, for a single downstream task, in memory limited settings. ROSA excels in such settings as it retains the low-memory-usage property of LoRA while achieving significantly higher performance (on par with full fine-tuning) on the downstream task.
>
> **Response to question 1** -
> Indeed, starting fine-tuning from the pre-trained checkpoint is a beneficial feature of both ROSA and LoRA. However, our ablation study shows that initializing LoRA with SVD (cf row “ROSA (SVD Init)” in Table 4) leads to better performance on the CoLA dataset. This supports our intuition that leveraging the pre-trained features to initialize the adapter is beneficial, compared to initializing the adapter to zero and learning new features from scratch. We added a sentence clarifying this observation in the ablation study.
>
> **Response to question 2** -
> Indeed, LoRA assumes that the downstream fine-tuning task or domain adaptations can be achieved with a low-rank tuning. In contrast ROSA does not make this assumption and in Theorem 1 we show that ROSA can fine-tune a model to the same solutions as full fine-tuning. The synthetic experiment mainly serves as an illustration to demonstrate that the theoretical claims made for the ROSA method are indeed realized in practice (for non-linear models trained by gradient descent, in contrast to the linear regression model considered in Theorem 1). We realize we had not included the loss curve for full fine-tuning. We have included it to better communicate the intent of Figure 2. (black curve labelled FT in Figure 2).We have also added additional remarks in  Section 4.1 to better communicate this point.

---

> > ### Author Response · Authors · 2023-11-21
> > **Completed additional ablation study experiments (Last day for discussion period)**
> >
> > We have completed additional ablation study experiments (Table 4) and added experiments for QQP in Table 2 (the final rank=8 runs will be added to the camera ready version). To facilitate a discussion we encourage you to provide a response before the end of the discussion period (Nov. 22).

---

> > > ### Author Response · Authors · 2023-11-22
> > > **We kindly ask you to consider increasing your score, given that we addressed the following**
> > >
> > > We believe that we have addressed your concerns on
> > > 1. Adding experiments on the QQP dataset
> > > 2. Adding more ablation experiments
> > > 3. Demonstrating the benefit of initializing using SVD
> > > 4. Clarifying the intent of the synthetic experiments.
> > >
> > > In which case, we would kindly ask you to consider increasing your score accordingly

---

### Author Response · Authors · 2023-11-19
**General response to reviewers**

We would like to thank the reviewers for their thoughtful feedback. Based on comments of the reviewers we have addressed the following and made updates in the PDF file accordingly. All revisions are marked in blue text for clarity.

1. **Addressing Reviewer 1 “previous publication” point:** We have clarified that the previous ROSA paper was presented in a non-archival workshop, as well as less than 4 months from the submission deadline of ICLR. Therefore, from both the perspective of contemporaneous work and also of its previous appearance being non-archival, we are not required to compare our work to this work.
2. **Addressing Reviewer 1 “more experiments” point:** We have added experiments on the QQP dataset to Table 2 (rank=2 and rank=8 is marked in blue in the updated file)
3. **Addressing Reviewer 1 “intent of synthetic experiments” point:** We have clarified the intent of the synthetic experiments - They serve to demonstrate that ROSA can indeed find solutions as good as full fine-tuning in practice. Which aligns with our claim in Theorem 1.
4. **Addressing Reviewer 1 “more ablations” point:** We have added ablation results for STS-B as well (Table 4)
5. **Addressing Reviewer 2 “compare with full FT and layer-wise training” point:**  We have clarified that full fine-tuning experiments are present throughout the paper as a point of comparison against LoRA and ROSA, both in terms of performance and speed, the two dimensions mentioned by Reviewer 2. With regards to layer-wise training, we present the result of a small-scale experiment in our reply, and show that the latency makes this method prohibitively expensive and incomparable to PEFT methods.
6.  **Addressing Reviewer 2 “ROSA motivation” point:** We have rephrased some sentences in the introduction and methodology to emphasize that the motivation of ROSA is to reduce the memory usage during fine-tuning while retaining the same level of performance as full fine-tuning.

---

### Meta-Review · Area_Chair_GHBp · 2023-12-07

**Metareview:**

The paper proposes Random Orthogonal Subspace Adapter (ROSA), a parameter-efficient fine-tuning method that significantly outperforms previous techniques while maintaining zero latency overhead during inference, and demonstrates its effectiveness in natural language generation and understanding tasks with GPT-2 and RoBERTa models. After careful reading of the paper, reviews, and the rebuttals, I am leaning towards not to accept the paper of its current form due to the following concerns. First, the experiment setting is not that strong, the perfomance gain over RTE and CoLA doesn't sound too convincing to me as these have the smallest test set even in GLUE. This applies to the generation side of the experiment as well. Second, as a study that aims to beat the widely used LoRA method, only showing emperical results on BERT/GPT-2 size model is clearly not enough. I understand there are computational reasons of doing so, however, on the LoRA side, using LoRA to finetune a 7B size model doesn't require a computational resouce beyond reach. This can be a very practical restriction on widely usage of the method proposed here.

About the workshop paper confusion, I would say it is clearly a valid submission to the ICLR. However, I do think the citation section says : "@inproceedings{hameed2023rosa,
  title={ROSA: Random Orthogonal Subspace Adaptation},
  author={Marawan Gamal Abdel Hameed and Guillaume Rabusseau}
  maintitle = {International Conference on Machine Learning},
  booktitle = {Efficient Systems for Foundation Models},
  year={2023}
}" in the github did confuse me for a while until I did a careful search. This isn't the best practice in the field. (This doesn't affect my judgement on the paper. I just would like to raise the point to explain a bit why reviewers can feel that the paper has been published in ICML.)

**Justification For Why Not Higher Score:**

n/a

**Justification For Why Not Lower Score:**

n/a

---

### Decision · Program_Chairs · 2024-01-16

Reject